# Assessment of the Brazilian Version of the Epworth Sleepiness Scale Using Item Response Theory: A Psychometric Study

**Cleber Lopes Campelo** [1,2,*] , **Rosângela Fernandes Lucena Batista** [1] , **Victor Nogueira da Cruz Silveira** [1] , **Kely Nayara dos Reis Silva Figueiredo** [1] , **Patrícia Maria Abreu Machado** [3] , **Adriano Ferreti Borgatto** [4] and **Alcione Miranda dos Santos** [1]

1   Department of Public Health, Postgraduate Program in Collective Health, Federal University of Maranhão, São Luís 65020-070, MA, Brazil; rosangela.flb@ufma.br (R.F.L.B.); victor.ncs@discente.ufma.br (V.N.d.C.S.); kely.nayara@discente.ufma.br (K.N.d.R.S.F.); alcione.miranda@ufma.br (A.M.d.S.)
2   Higher School of Health Sciences, State University of Amazonas, Manaus 69065-001, AM, Brazil
3   Department of Medicine I, Directorate of Technologies in Education, Federal University of Maranhão, São Luís 65020-240, MA, Brazil; patricia.machado@ufma.br
4   Department of IT and Statistics, Postgraduate Program in Physical Education, Federal University of Santa Catarina, Florianópolis 88040-970, SC, Brazil; adriano.borgatto@ufsc.br
*   Correspondence: ccampelo@uea.edu.br

**Abstract:** There seems to be a consensus that adolescents worldwide are not getting enough sleep. This study aimed to evaluate the psychometric properties of the Epworth Sleepiness Scale (ESS) in adolescents using the item response theory. A psychometric study was conducted with 2206 adolescents aged 18 and 19 years in the city of São Luís, Maranhão, Brazil. The dimensionality of the ESS was assessed by principal component analysis. A Samejima's graded response model (SGRM) was fitted to it. The findings of this study showed a good internal consistency and the unidimensionality of the ESS. Considering the latent trait continuum, we obtained three levels, with anchor items. For the item 'Sitting still in a public place', the adolescents presented a small possibility of dozing in level 1, and a medium and great possibility in level 2. The item 'Sitting around talking to someone' presented small, medium, and great possibilities of dozing in level 3. The ESS with five items showed satisfactory psychometric properties. In addition, the results from the SGRM showed that adolescents with high levels of sleepiness are likely to nod off or sleep sitting up while talking to someone. This study allows us to understand excessive daytime sleepiness in adolescents.

**Keywords:** sleep; item response theory; surveys and questionnaires

## 1. Introduction

Excessive daytime sleepiness (EDS) is defined by the International Classification of Sleep Disorders as the inability to remain awake and alert during the main periods of the day, associated with an irrepressible need to sleep or fall asleep [1]. Insufficient sleep is its main cause, and this disorder has a reported prevalence rate of 10.4 to 45.6% [2]. There seems to be a consensus that adolescents around the world are not sleeping for enough hours, which is why EDS is common in this population [3].

Sleep can be measured through polysomnography, which is considered the gold standard test for assessing sleep disorders, through the maintenance of wakefulness test (MWT) which quantifies the tendency toward wakefulness, measuring the ability to remain awake during sleepy circumstances, and by actigraphy, an examination that involves the use of a small wristwatch-like device that can delineate sleep and wakefulness based on movement. However, measurement and evaluation instruments have been widely used in clinical practice and epidemiological studies to measure EDS due to their low cost and easy access compared to other tests [3,4]. Even with their widespread use, self-report instruments with which to investigate sleep disorders face methodological challenges

related to their psychometric properties, requiring further investigation to obtain more accurate inferences and avoid the appearance of measurement bias [5].

The most common way to assess this sleep disorder is through the Epworth Sleepiness Scale (ESS) instrument, based on observations related to the nature and occurrence of daytime sleepiness [6]. Composed of eight questions, their scores are added to obtain a single score of 0 to 24 points per individual, where a higher score represents greater EDS [7].

The classification of EDS based on the total score is given at the test level, where the behaviour of all items that compose an instrument is considered [8]. However, for the items in a scale to be summarized in a single score, the scale must be unidimensional; that is, only one latent trait should be measured by the scale items.

Although the ESS is used to assess a single latent trait (of EDS) and has been validated in different countries and populations, showing good reliability and sensitivity [9–11], recent studies have pointed out some doubts about the dimensionality and behaviour of the items [2,12,13]. Despite the disagreements about the dimensionality, the ESS is widely used in different populations; however, in some cases, the response to an item on the scale may have a different meaning according to the respondents' characteristics, so the studies that point to the multidimensionality of the ESS have suggested an analysis at the level of the items [13,14].

When using evaluation instruments for diagnostic purposes or in research, we must obtain the largest amount of information possible regarding their psychometric properties, strengths, and limitations for the correct interpretation of the results [15]. In this sense, the item response theory (IRT) allows for the measurement of an unobservable variable, through a set of responses at the level of the items. IRT should be understood as a set of psychometric models to develop and refine psychological measures; its statistical models are based on the probability of conditioned responses to the latent trait under investigation, allowing for a more accurate measurement [16].

Given the above, the main objective of this study is to evaluate the psychometric properties of the ESS through IRT, with an assessment of its dimensionality, item parameters, scale refinement, estimation of the latent trait, identification of anchor items, and creation of a numerical scale with which to assess the level of sleepiness in 18-and 19-year-old adolescents.

## 2. Results

In this study, data from 2206 adolescents were assessed. The results indicated a predominance of female adolescents in the study sample (52.1%), aged 18 years (69.2%), mostly students (83.4%), who reported the skin colour of brown/mulatto (62.5%) and were in economic class B (48.5%). Table 1 shows the distribution for each level of answers given to the ESS for the sample under study.

**Table 1.** Frequencies of absolute and relative responses for each item of the BR-ESS. Brazil, 2016.

| How Likely Are You to Nap or Sleep, and Not Just Feel Tired, in the Following Situations? | Never (0) N (%) | Small (1) N (%) | Average (2) N (%) | Large (3) N (%) |
|---|---|---|---|---|
| 1—Sitting and reading | 770 (34.90) | 813 (36.85) | 398 (18.04) | 225 (10.20) |
| 2—Watching TV | 468 (21.21) | 733 (33.23) | 528 (23.93) | 477 (21.62) |
| 3—Sitting still in a public place (for example, at a movie theatre, meeting, or lecture) | 1415 (64.14) | 495 (22.44) | 183 (8.30) | 113 (5.12) |
| 4—Riding in a car for an hour non-stop as a passenger. | 933 (42.29) | 566 (25.66) | 366 (16.59) | 341 (15.46) |
| 5—At bedtime in the afternoon when possible | 169 (7.66) | 350 (15.87) | 488 (22.12) | 1199 (54.35) |
| 6—Sitting down talking to someone | 1854 (84.04) | 296 (13.42) | 43 (1.95) | 13 (0.59) |
| 7—Sitting still after lunch, with no alcoholic beverage. | 679 (30.78) | 636 (28.83) | 430 (19.49) | 461 (20.90) |
| 8—In a car stopped in traffic for a few minutes | 1461 (66.23) | 470 (21.31) | 183 (8.30) | 92 (4.17) |

The dimensionality analysis of the ESS via PCA proved to be sufficiently unidimensional, explaining 34.12% of the variance of the responses to the items, thus confirming its unidimensionality. The analysis of the internal consistency of the ESS showed a Cronbach's alpha coefficient ($\alpha$) of 0.71 and McDonald's omega of 0.76, confirming the high reliability of the original version.

The SGRM, according to the criteria of lower AIC and lower BIC, was the model chosen for the latent trait estimation, also presenting a good fit compared to other IRT models for polytomous items (Table 2). However, items 2 ("Watching TV"), 5 ("Going to bed in the afternoon when possible"), and 7 ("Sitting still, quiet after lunch, without drinking alcohol") showed a discrimination power below 0.65. Therefore, these items were removed from the latent trait estimation, and a new recalibration of the SGRM was performed containing only five items.

**Table 2.** The goodness of fit and adjustments of the IRT models for the BR-ESS. Brazil, 2016.

| Template Information Criteria | Eight Items | | | Five Items |
| --- | --- | --- | --- | --- |
| | Samejima's Graded Response Model | Generalized Partial Credit Model | Rasch Model for Polytomous Items | Samejima's Graded Response Model |
| AIC (Akaike's information criterion) | 37,201.85 | 37,344.27 | 37,620.81 | 20,904.60 |
| BIC (Bayesian information criterion | 37,384.21 | 37,526.64 | 37,683.50 | 21,018.58 |
| Goodness of fit | | | | |
| RMSEA (root mean square error of approximation) | 0.06 | 0.06 | 0.05 | 0.06 |
| SRMR (standardized root mean square residual) | 0.04 | 0.03 | - | 0.03 |
| CFI (comparative fit index) | 0.94 | 0.95 | 0.69 | 0.98 |

For the ESS with five items, the unidimensionality was maintained, with increased variance, explaining 43.31% of the answers, and the model presented a good fit (CFI = 0.97; TLI = 0.95; RMSEA = 0.06 (95% CI: 0.04–0.07), and SRMSR of 0.03). The Cronbach's alpha coefficient presented a value of 0.66 and the McDonald's omega assumed a value of 0.7, maintaining good reliability.

Table 3 presents the parameter estimates of the items on a scale (0,1), i.e., mean zero and standard deviation one, for the five items of the final model. We can observe that all five items showed good discrimination parameters (>0.65), with the "In a car, stopped in traffic for a few minutes" item being the one that best distinguishes EDS among adolescents (1.732).

**Table 3.** Factor loadings and item parameters (discrimination ($a_i$) and position ($b_{ik}$)) of the SGRM model of the BR-ESS. Brazil, 2016.

| How Likely Are You to Nap or Sleep, and Not Just Feel Tired, in the Following Situations? | Factor Loadings | $a_i$ | se ($a_i$) | $b_1$ | se ($b_1$) | b2 | se ($b_2$) | b3 | se ($b_3$) |
| --- | --- | --- | --- | --- | --- | --- | --- | --- | --- |
| Sitting and reading | 0.62 | 0.71 | 0.05 | −0.35 | 0.08 | 1.27 | 0.11 | 1.56 | 0.13 |
| Sitting still in a public place (for example, at a cinema, meeting, or lecture) | 0.64 | 1.11 | 0.10 | 0.96 | 0.09 | 1.58 | 0.10 | 1.67 | 0.12 |
| Riding in a car for an hour non-stop as a passenger | 0.62 | 0.69 | 0.05 | 0.47 | 0.10 | 0.87 | 0.11 | 0.80 | 0.12 |
| Sitting around talking to someone | 0.53 | 1.05 | 0.10 | 2.03 | 0.16 | 2.89 | 0.20 | 2.84 | 0.32 |
| In a car stopped in traffic for a few minutes | 0.65 | 1.13 | 0.10 | 1.05 | 0.10 | 1.56 | 0.10 | 1.89 | 0.13 |

Regarding the parameter of position, the item "Sitting and reading" presented the lowest values in the first category ($b_1$), and "Riding in a car for an hour non-stop as a passenger" in the second and third categories ($b_2$ and $b_3$). The highest parameters of position in all categories belonged to the item "Sitting around talking to someone" (Table 3). Therefore, adolescents with lower latent trait values (lower level of diurnal sleepiness) easily endorsed the first two items, while only those with higher latent trait values could endorse the latter.

Considering the linear transformation of the latent trait for the scale (30,10), three anchor levels, obtained from the mean ± one standard deviation, were considered: level 40, level 50, and level 60 (Table 4). Only three levels presented anchor items, with level 40 being the adolescents who presented a small possibility of dozing while sitting and reading (item 1). Item 2 (sitting and talking to someone) concentrated all response categories at level 60, which is similar to item 3 (sitting still in a public place and in a car stopped in traffic for a few minutes), which concentrated all response categories at level 50.

**Table 4.** Anchor item positions in the continuum.

| Items/Latent Trait | 0 | 10 | 20 | 30 | 40 | 50 | | 60 | | |
|---|---|---|---|---|---|---|---|---|---|---|
| Item 1 | | | | | 0 | 1 | 2 | | | |
| Item 2 | | | | | | | | 0 | 1 | 2 |
| Item 3 | | | | | 0 | 1 | 2 | | | |

0, response category: small possibility of napping; 1, response category: medium possibility of napping; 2, response category: great possibility of napping.

## 3. Discussion

The results obtained indicate that the BR-ESS has psychometric properties suitable for use with 18- and 19-year-old adolescents. IRT allowed the identification of the most relevant items in the context of sleepiness for 18- and 19-year-old adolescents, as well as the construction of a numerical scale with which to measure the intensity of EDS and identify adolescent behaviours that directly influence this sleep disorder. Another important finding is that the scale was built without including items 2, 5, and 7 of the ESS, since these items showed a low discriminative power in this population.

Some studies [13,14] that assessed the dimensionality of the ESS suggest that it provides a reliable measure of one's propensity to sleepiness, but presents different constructs, suggesting that the instrument analyzes two constructs related to EDS. In this study, the SSE showed good reliability and a good fit to the unidimensional model, both in its full version and in the scale proposed in this study, when measuring only a single latent trait.

On the other hand, recent studies [2,12] question these assessments and, using different statistical methods, confirm their unidimensionality. Given this, the same authors also suggest a further clarification on the behaviour of the items that make up the ESS. Thus, the model adjusted by TRI in this study enabled the identification of a suitable model for understanding this phenomenon in a population of adolescents.

Regarding the levels of the measurement scale, it was observed that the highest level (level 40) of the scale proposed in this study was composed of items that presented the greatest power of discrimination; thus, this level is composed of adolescents who, besides dozing while "sitting and reading" or "sitting and talking to someone", also have a propensity for sleepiness when they are "sitting still in a public place" or "in a car stopped in traffic for a few minutes".

Sleep disorders such as EDS have as their primary cause sleep deprivation and are associated with diurnal alterations with a decrease in attention [4]. There is evidence that, when individuals are prevented from acquiring sufficient hours of sleep, they tend to nap in quiet environments during the day [3]. This high power of discrimination of item 3 ("Sitting still in a public place") can be explained by the fact that it refers to a situation with a great capacity to cause drowsiness.

In this sense, the literature shows that adolescents with sleep deprivation tend to nap during classes, and this is the cause of greater difficulties in attention/concentration and lower school performance among adolescents [17]. We emphasize that, although the item refers to places such as a cinema, meeting, or lecture, the daily life of adolescents is marked by more time spent in the classroom, a public place similar to the situations referred to in the item.

An interesting finding of this study is the high discrimination power of item 8 ("In a car stopped in traffic for a few minutes"). A study that assessed the ESS among college

adolescents indicates that this item is probably measuring a construct related to sleepiness, different from the other items, and suggests that it can be excluded when calculating the total score [14]. In our sample, this item obtained a high discriminatory power.

A systematic review study [4] on subjective sleep assessment instruments points out that some authors used a modified version of the ESS for this item when measuring EDS in adolescents aged 10 to 16 years, replacing it with "doing homework/taking a test"; however, despite good reliability, this modified version did not undergo construct validation.

Items 2 ("Watching TV"), 5 ("Going to bed in the afternoon when possible"), and 7 ("Sitting still after lunch, with no alcohol") showed a low discrimination power for EDS; therefore, they were excluded during the estimation of the latent trait and construction of the interpretable numerical scale.

Adolescents lead the ranks of mobile phone and internet use in Brazil, and this probably decreases the time of TV use [18]. This result reflects the change in behaviour among adolescents who grew up in the technological and digital era, as well as the constant changes and innovations of electronic devices.

Despite the low discrimination power and the fact that it was not considered in the estimation of EDS intensity in this study, the use of television as well as the indiscriminate use of electronic devices has been pointed out as having harmful consequences for sleep. According to a systematic review, the use of TV, the internet, and electronic games is associated with a delay in bedtime, causing unfavourable outcomes such as EDS [19].

The other items that showed less discrimination power refer to the propensity to sleepiness, i.e., to go to bed in the afternoon or after lunch when possible. The literature shows that morning school hours play a significant role in reducing the hours of sleep of adolescents; those who have later school start times report more hours of sleep, better sleep, and lower EDS when compared to students who start school earlier in the day [3,17]. The low discrimination of this item is probably related to the school starting time, because adolescents who study in the afternoon or have extracurricular activities at this time do not usually go to bed in the afternoon.

Regarding the difficulty of the items of the ESS, item 6 ("Sitting talking to someone") presented the greatest difficulty, because this item refers to a situation where dozing is less likely. Loss of interest in daily activities, behavioural disorders, and poor social relationships, as well as attention and concentration disorders, are perceived in individuals with more advanced degrees of EDS [20]. Thus, this item was answered positively by adolescents with a greater intensity of sleepiness.

Modelling via IRT enabled the construction of an interpretable numerical scale with which to assess sleepiness in adolescents aged 18 and 19 years, allowing a quantitative analysis and qualitative interpretation regarding sleep that are aligned with reality, since the ESS was built and validated in 1991 [6], in a context of globalization different from that of the present day.

No scale for sleepiness built based on IRT was found in the literature; therefore, the results obtained here were interpreted considering the estimation of the latent trait and the process of the construction of scales via IRT, known as the anchoring of the items [16].

As a strong point of this study, we highlight the use of IRT as an analysis tool that allowed us to fill the gaps in the dimensionality and behaviour of the ESS items, and contributes to the understanding of EDS in adolescents, because it goes beyond the commonly assigned answers in sleep assessment instruments, allowing us to analyze how much the item discriminates the latent trait and information for the measure. Such findings would not be possible in analyses performed using the classical theory of tests.

However, the present study also has its limitations. Although we used a large sample, it is important to highlight that the study was carried out exclusively in the Brazilian population. Another limitation refers to the age of the adolescents; all were between 18 and 19 years old, so the results found here do not apply to younger adolescents. For future studies, we suggest a more heterogeneous sample of adolescents in terms of age to verify whether the psychometric properties of the ESS remain adequate.

## 4. Materials and Methods

This is a psychometric study nested in a prospective cohort of adolescents, conducted in the city of São Luís, Maranhão. The data used come from an open cohort that uses data from individuals born between July 1997 and June 1998 and who were aged between 18 and 19 between July 2015 and June 2016.

The first phase of the cohort was conducted in ten public and private maternity hospitals; the sample of 2542 births consists of one-third of the births that occurred in that year. The second phase occurred in 2005 and 2006, totalling 673 children [21].

The sample that makes up the third phase of this cohort is composed of adolescents aged 18 to 19 years, assessed in 2016, comprising 684 adolescents. To increase the power of the analyses and avoid future losses, the cohort was opened to include other individuals born in São Luís, Maranhão, in 1997. The search for adolescents was carried out using the Information System on Live Births (SINASC) database as the first step, with an inclusion criterion of being born in a maternity or hospital located in São Luís–MA in 1997. From this list, a random raffle was carried out, resulting in a total of 4593 births in 1997. Of these, 1133 adolescents were contacted by phone or in person. Later, volunteers were identified through schools, universities, and social networks. In this step, 1831 individuals were included, totalling 2515 adolescents. They were subjected to the same tests and questionnaires as the others. In addition, a questionnaire was applied to the mothers of these adolescents to retrospectively collect perinatal data. The methods used in the birth cohort are detailed in Simões et al. (2020) [22].

Of the 2515 adolescents in the third phase of the cohort, 309 were excluded from this study because they did not fully respond to the Brazilian version of the Epworth Sleep-iness Scale (BR-ESS) instrument and/or because they did not present all the variables of interest to the study, thus resulting in a total of 2206 adolescents in the sample.

The BR-ESS is a self-reported questionnaire, developed by Johns (1991) [5] and trans-lated and validated by Bertolazi et al. [9], used to assess the level of excessive daytime sleepiness through a four-point adjectival scale, where "0" indicates never napping, "1" a small possibility of napping, and "2" a medium possibility of napping, while "3" indicates a high probability of napping. The scores of the items are added to obtain a total score, which ranges from 0 to 24 points; the higher the score, the higher the level of sleepiness, and scores above 10 already indicate EDS [23].

To characterize the socio-demographic and economic information of the study sample, the following variables were assessed: gender (male and female), age (in years), skin colour (white, black/black, brown, yellow/orange, or indigenous), studies or works (yes or no), and socioeconomic stratum (A, B1, B2, C1, C2, or D/E class), according to the criteria of Brazil Economic Classification/BEC [24]. All variables assessed in this study were obtained from the research instruments used in the third stage of the cohort.

Statistical analyses were performed using R and RStudio. Categorical variables are expressed as absolute values and/or percentages. The dimensionality of the ESS was assessed by principal component analysis (PCA) from the polychoric correlation matrix and using the ScreePlot plot, following the Reckase criterion (1990) [25], which considers the ESS to have a dominant unidimensionality when the first given value corresponds to at least 20% of the total variance. These analyses were performed via the psych package of the R software [26].

After confirming the unidimensionality of the ESS, the internal consistency of the items that make up the scale was measured by Cronbach's alpha and McDonald's omega coefficients, which accurately estimate the reliability of the instrument. The interpretation of the two tests occurs similarly; the closer to one, the more reliable the test. For this study, the following were adopted as reference values: <0.6, low reliability; between 0.6 and 0.7, moderate reliability; and above 0.7, high reliability [27].

The analysis of the psychometric properties of the ESS was conducted based on the IRT. Three unidimensional models for polytomous items were fitted: Samejima's graded response model (SGMR); generalized partial credit model (GPCM); and Rasch model for

polytomous items (RM) [28]. The model with the best fit was used to estimate the latent trait, refine the scale considering item discrimination, identify anchor items, and create an interpretable numerical scale from the latent trait, to measure the ESS in adolescents aged 18 and 19 years.

The parameters of interest to models described above are the parameter of item discrimination ($a_i$) and $b_{ik}$, the parameter of difficulty or position of the category k of item i. The parameter $a_i$ refers to the ability of an item to differentiate the intensity of sleepiness among adolescents, with the expectation that items with good discrimination have ai ≥ 0.65. The parameter $b_{ik}$ represents the difficulty of a given category of the item; the higher its value, the greater the difficulty of the item [28]. This study represents the position on the latent trait scale where there is a probability of 50% or greater of endorsement from one response category to another.

The estimation of the parameters of the IRT one-dimensional models was performed using the Mirt package from R [29]. For the selection of the model among those that were adjusted to estimate the latent trait, the AIC (Akaike information criterion) and BIC (Bayesian information criterion) were used to compare the models; lower values indicate a better fit [30]. The quality of the model adjustment (goodness of fit) was evaluated using the following indices: for the RMSEA (root mean square error of approximation), values lower than 0.08 indicate adequate adjustment, and lower than 0.06 a good adjustment; for the SRMR (standardized root mean square residual), values lower than 0.05 were considered; and for the CFI (comparative fit index) and TLI (Tucker–Lewis Index), values higher than 0.95 were considered [30,31].

In IRT, the data are evaluated assuming that the latent trait is measured on a scale such that, for the population assessed, the mean is zero and the standard deviation equals one, thus being theoretically able to assume any real value between $-\infty$ and $+\infty$ [24]. Only items with good discrimination remained in the model with item refinement, indicating that they are associated with the latent trait under investigation [16]. After the estimation of the latent trait, the construction of the numerical scale was performed, based on the anchor levels to measure EDS in adolescents aged 18 and 19 years old.

For an item to be considered an anchor at a given level of the scale, it is expected that it has a discrimination level equal to or greater than one, and that some category of this item is chosen by at least 65% of respondents with the same level of the latent trait; in addition, the difference between the response probabilities between the item and the subsequent level must be at least 30% [16,28].

Initially, the latent trait values were estimated on a metric scale with a mean of 0 and a standard deviation of 1. This technique aims to facilitate the computational process for estimating the parameters of the model [16]; however, to facilitate interpretation, a linear transformation was performed for 30,10, i.e., mean 30 and standard deviation 10. The anchor items and their content were analyzed in the latent trait scale and their interpretation was performed based on the theme under study.

The study was approved by the Research Ethics Committee of the University Hospital of the Federal University of Maranhão (ruling number 1,302,489 and CAEE number 49096315.2.0000.5086).

## 5. Conclusions

The ESS scale with five items has adequate psychometric properties when measuring a single latent trait. In addition, the results from the SGRM showed that item 6 of the ESS showed greater difficulty, reflecting that adolescents with high sleepiness are likely to nod off or sleep sitting up while talking to someone. It was also possible to observe three levels of excessive daytime sleepiness among the adolescents under study, allowing us to understand the behaviour of these adolescents with regard to excessive daytime sleepiness.

Our study was restricted to adolescents aged 18 and 19 years, so we ask readers to be careful when generalizing the results to adolescents in general; therefore, we are suggesting

further studies that evaluate the behaviours of the ESS items in other populations of adolescents with different age groups.

**Author Contributions:** C.L.C., R.F.L.B. and A.M.d.S. designed the research; C.L.C., R.F.L.B., A.M.d.S. and P.M.A.M. conducted the research; C.L.C., V.N.d.C.S., P.M.A.M. and A.M.d.S., A.F.B. analyzed data; and C.L.C., R.F.L.B., V.N.d.C.S., K.N.d.R.S.F., P.M.A.M. and A.M.d.S. wrote the paper. C.L.C. held primary responsibility for the final content. All authors have read and agreed to the published version of the manuscript.

**Funding:** This research was funded by Departamento de Ciência e Tecnologia (DECIT) of the Ministério da Saúde through a grant from the Conselho Nacional de Desenvolvimento Científico e Tecnológico (CNPq), grant number 400943/2013-1. JRC receives a grant from the Fundação de Amparo à Pesquisa e ao Desenvolvimento Científico e Tecnológico do Maranhão (FAPEMA), grant number BIC-01991/20. The APC was funded by Fundação Josué Montello of the Universidade Federal do Maranhão, process number 17617/2017-29.

**Institutional Review Board Statement:** The study was carried out following the recommendations of the Declaration of Helsinki and was approved by the Research Ethics Committee of the University Hospital–Federal University of Maranhão (UFMA), under Opinion no. 1.302.489. In all phases of the cohort, the Informed Consent Form was signed by the individual or guardian. All projects meet the criteria in Resolution (466/2012) of the National Health Council and its complementary regulations.

**Informed Consent Statement:** Informed consent was obtained from all subjects involved in the study. Written informed consent has been obtained from the patient(s) to publish this paper.

**Data Availability Statement:** The data are unavailable due to privacy concerns.

**Acknowledgments:** We would like to thank all the institutions that contributed to the financing, preparation, and execution of this study and we thank the adolescents involved in the research and the funding agencies.

**Conflicts of Interest:** The authors declare no conflict of interest.

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
