# Peer review of "Assessment of the Brazilian Version of the Epworth Sleepiness Scale Using Item Response Theory: A Psychometric Study"

_2624-5175, doi:10.3390/clockssleep5040038_

Round 1
Reviewer 1 Report
Manuscript ID: clockssleep-2546216
Type of manuscript: Article
Title: ASSESSMENT OF THE BRAZILIAN VERSION OF THE EPWORTH SLEEPINESS SCALE USING ITEM RESPONSE THEORY: A PSYCHOMETRIC STUDY
This is a study of assessment on the Epworth sleepiness scale using the item response theory specially for the adolescents aged 18-19 years. This paper is well-written. However, I have few comments to improve the paper.
1.Table 1: N (%) are presented. Item 3 and 6 and 8, N values are given as decimal with places. My understand is that is a mistake and should be presented as 1415 instead of 1.415 for item 1.
2. In the same Table 1, item 8 "Never" N value is wrong (1.46). It should be 1461.
3. Items are selected based on the discrimination power. I do not see any discrimination power table for items. Need to include that.
4. Comparing the dimensionality analysis based on items 8 and 5 do not show very big improvement as variation with <10% increase and similar Cronbach's alpha and McDonald's Omega values. IRT models provide the further improvement with five items. Could you please provide the goodness of statistics for models with eight items?
5. In the methods section- first paragraph mentioned that " The study only included participants from the third phase of the cohort". Third paragraph mentioned that third phase consist of only 684 adolescents. Later on the third paragraph you mentioned that this is a open cohort. These statements are confusing to the readers. Better to rewrite to mentioned that this is a open cohort including 2016 assessed adolescents 18-19 years. Also, I have a problem of taking only 1997 born individuals will be 19 years by 2016. How about 1998 born individuals will be 18 years by 2016. Any reasons to select the 1997 year?
I think adding the information of answering these questions will improve the paper.
Author Response
Initially, we would like to thank the considerations and time spent by reviewer #1
This is a study of assessment on the Epworth sleepiness scale using the item response theory specially for the adolescents aged 18-19 years. This paper is well-written. However, I have few comments to improve the paper.
Comment #1
1.Table 1: N (%) are presented. Item 3 and 6 and 8, N values are given as decimal with places. My understand is that is a mistake and should be presented as 1415 instead of 1.415 for item 1.
We apologize for this error. Proper punctuation would be to use a comma to indicate a value in thousands. However, in service of the reviewer, we will leave the value in full.
- In the same Table 1, item 8 "Never" N value is wrong (1.46). It should be 1461.
The same procedure used in the previous comment was performed. It was supposed to be a number in the thousands, but we have already corrected the entire table 1 with integer values.
- Items are selected based on the discrimination power. I do not see any discrimination power table for items. Need to include that.
We made a correction in the title of Table 3, indicating the abbreviations of the discrimination and position parameters.
- Comparing the dimensionality analysis based on items 8 and 5 do not show very big improvement as variation with <10% increase and similar Cronbach's alpha and McDonald's Omega values. IRT models provide the further improvement with five items. Could you please provide the goodness of statistics for models with eight items?
We revised the table to describe the five-item and eight-item models and make the process of deciding which model to use even clearer.
- In the methods section- first paragraph mentioned that " The study onlyincluded participants from the third phase of the cohort". Third paragraph mentioned that third phase consist of only 684 adolescents. Later on the third paragraph you mentioned that this is a open cohort. These statements are confusing to the readers. Better to rewrite to mentioned that this is a open cohort including 2016 assessed adolescents 18-19 years. Also, I have a problem of taking only 1997 born individuals will be 19 years by 2016. How about 1998 born individuals will be 18 years by 2016. Any reasons to select the 1997 year?
We apologize for the misinterpreted text in this section. We corrected it in as much detail as possible to make it understandable and avoid further confusion.
Reviewer 2 Report
Compelo an coauthors show clear focussed data about a brazilian version of the ess. The data aquisition and its analysis is well done, the manuscript ist well written. No further comments.
Author Response
We would like to thank the considerations and time spent by reviewer #2
Reviewer 3 Report
The manuscript deals with a topic of greater interest and debate: the methods of assessment of EDS. Moreover, its implementation in a population of adolescents, in whom insufficient sleep is a common phenomenon, gives this study greater interest. The method is fair, with a sample size. Good introduction and discussion. Here are some suggestions for changes:
1.Introduction page 1, line 29: reference #1, maybe it should be replaced by the newest ICSD-3 TR 2023;
2.Introduction page 1, lines 29-31: expand epidemiological data also at the numerical level;
3.Introduction page 1, lines 32-34: specify that for the objective evaluation of EDS we can use tests such as the MWT (Maintenance of Wakefulness Test) and actigraphy for the evaluation of sleep deprivation, specifying briefly what they are;
4.Reference #13: it should be removed or better explained because it refers to college students, with an average age of 21-22 years.
5.Methods: Did all enlisted teenagers drive?
6.What does the acronym "SDE" stand for?
7.A limitation of the study is that it was conducted solely on the Brazilian population, although a plus point is the fact that it was a multicenter study. Another limitation of the study, rightly pointed out by the authors, is the enrollment of adolescents aged between 18 and 19 years, leaving out the other age groups.
8.As part of the conclusions, please expand to specify that it would be important to evaluate different items of EDS, depending on the adolescent age groups
Author Response
First, we wish to thank reviewer #3 for his valuable considerations.
The manuscript deals with a topic of greater interest and debate: the methods of assessment of EDS. Moreover, its implementation in a population of adolescents, in whom insufficient sleep is a common phenomenon, gives this study greater interest. The method is fair, with a sample size. Good introduction and discussion. Here are some suggestions for changes:
1.Introduction page 1, line 29: reference #1, maybe it should be replaced by the newest ICSD-3 TR 2023;
We evaluated the reference and included the reference in the manuscript.
2.Introduction page 1, lines 29-31: expand epidemiological data also at the numerical level;
We evaluated the suggestion and expanded the epidemiological data at numerical level as suggested, for this purpose we included a reference that was already included in the first version, however this changed the order of the references.
3.Introduction page 1, lines 32-34: specify that for the objective evaluation of EDS we can use tests such as the MWT (Maintenance of Wakefulness Test) and actigraphy for the evaluation of sleep deprivation, specifying briefly what they are;
We evaluated the suggestion, made the expansion, as suggested by the reviewer.
4.Reference #13: it should be removed or better explained because it refers to college students, with an average age of 21-22 years.
We thank you for the suggestion and decided to keep the reference in the text, although this study was carried out with university students aged 21 to 22 years, we believe that for cultural reasons they are similar to our research subjects, in addition this study was also important for pointing out some weaknesses of the ESS and suggesting analyzes such as those made in our article, served as a driving force so that we could question ourselves about the behavior at the level of the ESS items in adolescents
5.Methods: Did all enlisted teenagers drive?
No. This data was not measured in the cohort, what we have is an ESS item that asks about drowsiness in a car stopped in traffic, but does not specify whether as a driver or passenger, so we cannot say that the adolescents drove.
6.What does the acronym "SDE" stand for?
We apologize for this error, this abbreviation has been corrected. It was the EDS.
7.A limitation of the study is that it was conducted solely on the Brazilian population, although a plus point is the fact that it was a multicenter study. Another limitation of the study, rightly pointed out by the authors, is the enrollment of adolescents aged between 18 and 19 years, leaving out the other age groups.
We appreciate the suggestions regarding the limitations of the study and they were included in the article.
8.As part of the conclusions, please expand to specify that it would be important to evaluate different items of EDS, depending on the adolescent age groups
We appreciate the suggestion and it has been included in the conclusion of the article.
Round 2
Reviewer 1 Report
"Table 1, item 8 "Never" N value is wrong (1.46). It should be 1461."
This item is not corrected yet. It is not adding up to 2206.
In the methods section, still there is no mentioned of 2206 total.
Author Response
Response to Reviewer 1 Comments
First of all, we would like to thank reviewer 1 for his time and corrections, they contributed a lot to the improvement of the article.
Point 1: "Table 1, item 8 "Never" The value of N is wrong (1.46). It should be 1461."This item has not yet been corrected. It's not adding up to 2,206.
Response: We apologize for the mistake, we did what was requested and we took advantage of the opportunity and also corrected item 5 which had the same error.
Point 2: "In the methods section, there is still no mention of 2,206 in total."
Response: Thanks for the suggestion, we changed the method.
Reviewer 3 Report
The authors have addressed my suggestions
Author Response
First of all, we would like to thank the reviewer for his time and corrections, they contributed a lot to the improvement of the article. As there were no notes, I leave our gratitude.